# Selective Functionalization of High-Resolution Cu_2_O Nanopatterns via Galvanic Replacement for Highly Enhanced Gas Sensing Performance

**DOI:** 10.3390/s18124438

**Published:** 2018-12-15

**Authors:** Ju Ye Kim, Soo-Yeon Cho, Hee-Tae Jung

**Affiliations:** 1Department of Chemical and Biomolecular Engineering (BK-21 Plus), Korea Advanced Institute of Science and Technology (KAIST), 291 Daehak-ro, Yuseong-gu, Daejeon 34141, Korea; juyekim@kaist.ac.kr (J.Y.K.); chosooyeon@kaist.ac.kr (S.-Y.C.); 2Korea Advanced Institute of Science and Technology (KAIST) Institute for NanoCentury, 291 Daehak-ro, Yuseong-gu, Daejeon 34141, Korea

**Keywords:** gas sensor, nanopattern, chemical sensitization, galvanic replacement, p-type metal oxide, high-resolution

## Abstract

Recently, high-resolution patterned metal oxide semiconductors (MOS) have gained considerable attention for enhanced gas sensing performance due to their polycrystalline nature, ultrasmall grain size (~5 nm), patternable properties, and high surface-to-volume ratio. Herein, we significantly enhanced the sensing performance of that patterned MOS by galvanic replacement, which allows for selective functionalization on ultrathin Cu_2_O nanopatterns. Based on the reduction potential energy difference between the base channel material (Cu_2_O) and the decorated metal ion (Pt^2+^), Pt could be selectively and precisely decorated onto the desired area of the Cu_2_O nanochannel array. Overall, the Pt-decorated Cu_2_O exhibited 11-fold higher NO_2_ (100 ppm) sensing sensitivity as compared to the non-decorated sensing channel, the while the channel device with excessive Pt doping showed complete loss of sensing properties.

## 1. Introduction

Metal oxide semiconductors (MOSs) (e.g., n-type [1,2,3,4,5] and p-type [6,7,8,9,10,11,12]) are widely used as gas sensing materials due to their high sensitivity, large specific surface area, applicability to various gases (NO_2_, CO_2_, H_2_, volatile organic compounds (VOCs), etc.), high electron mobility, and good chemical/thermal stability at high operating temperature [13,14,15,16,17,18,19,20]. Many approaches have been adopted to enhance the gas sensing performance of the MOS, including grain size refinement, increasing the surface area, reducing the interface area between the substrate and the device, and introduction of dopants and defects [1,18,19,20,21,22]. Doping with noble metals such as Pt [21,22], Pd [23], and Ag [2] has been widely used to enhance the sensing performance of the MOS by tuning the gas adsorption or diffusion properties. As sensitizers on the MOS surface, noble metals can induce electronic or chemical sensitization pathways [1,21]. Electronic sensitization occurs due to the electronic interaction between the semiconductor and the doped metal. When metal dopants undergo partial oxidation in air, their oxidation state of the metal additive can be changed into metallic state depending on the environment. This increases the depth of the electron-depleted charge layer due to electron extraction from the metal oxide, thereby changing the electronic state change of the MOS. The chemical sensitization represents a spill-over effect by activating target gas adsorption on the semiconductor surface [24]. The doped promoter facilitates target gas as active state, leading to increases gas concentration on the channel surface. During the sensing process, the concentration of the adsorbed target gas is effectively increased by the promoter. 

Recently, a patterned p-type polycrystalline MOS with an ultrathin and a high aspect ratio showed 5-fold enhanced gas sensing performance due to its ultra-small grains (~5 nm) and large surface-to-volume ratio [6,25]. One of the promising ways to improve the sensing properties is noble metal decoration on the MOS sensing channel. The conventional methods for metal functionalization include sputtering [2,23], thermal evaporation [3], electrospinning [21], and wet impregnation [22] in an aqueous solution. Since the evaporation technique does not allow for selective functionalization of the MOS channel with noble metals, undesired decoration of noble metals on the electrode and substrate can hinder effective control of the doping concentration and cause non-uniform deposition. Moreover, non-directional decoration can produce another dopant-connected channel that reduces the sensitivity of the MOS channel and causes malfunction of the sensing device. Thus, for reduced channel dimensions and higher resolution, more selective and direct functionalization is required [26]. 

In this study, we selectively functionalized a high-resolution p-type Cu_2_O sensing channel with Pt nanoparticles by galvanic replacement in order to significantly enhance the gas sensing performance. High-resolution (10 nm scale) and high-aspect-ratio (~25) Cu_2_O nanopatterns are fabricated via a unique lithographic technique with low-energy plasma bombardments. The unique morphological characteristics (resolution, aspect ratio, grain composition, fully exposed structure) of the Cu_2_O nanopattern and the possibility of large-area fabrication make it the optimum nanostructure for the sensing channel of gas sensors. Galvanic replacement was employed to further enhance the gas sensing performance of the high-resolution Cu_2_O nanopattern. This facile decoration process involves a chemical reaction between two materials with different reduction potentials [27,28,29,30]. Hence, selective functionalization on the target site is possible, and the size and the amount of the state of Cu_2_O and to confirm the existence of Pt on the surface. Decorated dopants can be controlled by adjusting the precursor concentration and reaction conditions. The optimally functionalized nanochannel array (Cu_2_O/Pt) shows 11-fold higher sensitivity as compared to the pristine Cu_2_O array upon exposure to 100 ppm NO_2_ gas (300 °C), because of electronic and chemical sensitization effects of the decorated Pt nanoparticles.

## 2. Materials and Methods

### 2.1. Cu_2_O Nanopattern Fabrication

A polydimethylsiloxane (PDMS) mold was obtained by using a Si pre-patterned mask after curing a silicone elastomer mixture (Sylgard 184, 10:1 weight ratio of base to curing agent; Dow Corning, Midland, MI, USA). Polystyrene (18000 g/mol) solution in toluene was spin-coated (3000 rpm, 45 s) onto a SiO_2_-coated (200 nm) Si wafer to obtain a thin polystyrene (PS) film. The PS pre-pattern (500 nm width, 350 nm height) was created using capillary forces between PS and PDMS upon heating above the T_g_ (glass transition temperature, ~135 °C) in a vacuum oven. Then, the target metal (Cu, 25 nm thick) was uniformly deposited onto the PS pre-pattern by electron beam evaporation. By Ar ion bombardment at low energy (500 eV) with a wide angle distribution, the Cu thin layer was etched and emitted into the PS pre-patterned side wall to obtain a polycrystalline feature. After the PS residue was removed by oxygen reactive ion etching (RIE) under a low vacuum and O_2_ (100 standard cubic centimeters per minute (sccm)) plasma environment, thermal oxidation was performed in a tubular furnace at 450 °C for 3 h to obtain a Cu_2_O line channel.

### 2.2. Galvanic Reaction

To decorate Pt nanoparticles selectively onto the Cu_2_O line pattern, 1 mL of 1 mM potassium tetrachloroplatinate (K_2_PtCl_6_, Sigma Aldrich, Burlington, MA, USA), 1 mL of 1 mM hydrochloric acid (HCl, Sigma Aldrich), 5 mL of 20 mM sodium hydroxide powder (NaOH, Daejung), and 5 mL of 10 mM ascorbic acid (AA, Sigma Aldrich) were added to the Cu_2_O line pattern substrate. This reaction was performed at room temperature (RT) to control the reaction pace preventing over-loaded Pt. Then, the sample and solutions were injected into a 30 mL vial and agitated on a SK-300 benchtop shaker (90 rpm, Lab Companion, Daejeon, Korea). After each reaction, the substrate was cleaned with ethanol and DI water, and then dried under N_2_.

### 2.3. Characterization

The surface morphology of the fabricated Cu_2_O polycrystalline array and Pt-decorated line pattern was characterized by field-emission scanning electron microscopy (Magellan 400, Nova 230, ThermoFisher, Hillsboro, OR, USA) and energy-dispersive X-ray (EDS) spectroscopy (cmodel, manufacturer, city, xsttae abbrev if USA, country. The electron beam energy was 5 kV and 10 kV for the EDS analysis. For the line depth and width profiling of the metal oxide pattern array, atomic force microscopy (AFM; XE-100, Park Systems, Suwon, Korea) was used. X-ray photoelectron spectroscopy (XPS, K-alpha, ThermoFisher, Hillsboro, OR, USA) analysis was conducted to verify the oxidation state of Cu_2_O and to confirm the existence of Pt on the surface.

### 2.4. Sensor Fabrication and Measurement

To measure the resistance signal of the channel, 70-nm-thick Au electrodes with a predeposited 5-nm-thick Ti adhesion layer, as well as 100 μm spacing and width, were deposited the Cu_2_O-Pt line pattern by e-beam evaporation using a customized SERS mask [31]. For saturation of oxygen gas on the substrate, air (400 sccm) was injected over 3 h at 260 °C into the sealed gas reaction chamber where the prepared metal oxide channel device was located. The external sources are used to heat sensing substrate. Heater module is right below the chamber plate. The size of the sealed gas sensing chamber was approximately 10 cm (width) × 5 cm (height) × 8 mm (depth). The resistance signals were displayed directly on a computer via a customized data acquisition module (34970A, Agilent, Santa Clara, CA, USA) to verify the gas sensitivity. NO_2_ gas (100 ppm) was delivered into the sensing chamber every 10 min by a mass flow controller (MFC, 5850E, Brooks, Seattle, WA, USA). Air was used for observing the desorption properties.

## 3. Results and Discussion

### 3.1. Fabrication of Pt Decorated High-Resolution Cu_2_O Nanochannel

The overall fabrication of the Pt-functionalized Cu_2_O sensing channel (Cu_2_O/Pt) is illustrated in Figure 1. First, a PS prepattern with 500 nm width and 350 nm height was formed on a Si wafer substrate on which SiO2 was deposited via thermal pattern transfer using a PDMS mold (Figure 1a). By e-beam evaporation, a 25-nm-thick Cu nanofilm was uniformly deposited on the prepatterns (Figure 1b). The Cu layers were then etched such that they covered the side surfaces of the prepattern by using wide-angle distribution by a lower-energy plasma (500 eV, Ar^+^) bombardment process using ion-milling instruments. Bombardment of the Cu precursor layers with low-energy Ar^+^ plasma ruptures the Cu nanofilms to form 5-nm high-resolution grains that are sputtered over a wide area [32,33]. This leads to the formation of a polycrystalline Cu nanopattern with a high aspect ratio as a result of the attachment of the 5-nm Cu grains to the side wall of the PS prepatterns. After removing the polymer residue by oxygen RIE, ultra-high-resolution of the Cu line nanochannel remained (Figure 1d). To oxidize Cu into Cu_2_O, the fabricated nanopattern was subjected to thermal annealing in a tubular furnace at 450 °C for 3 h in air and then cooled to RT (Figure 1e). 

Since the galvanic reaction occurs only between target materials that have different reduction energy potentials, selective doping on the Cu_2_O device channel can be possible without undesired deposition (e.g., on the substrate). Functionalized Pt can enhance the sensing performance of the Cu_2_O channel due to the spill-over effect, i.e., easy dissociation of the adsorbed gas on the metal surface and subsequent migration into the metal oxide surface. Therefore, interparticle doping via galvanic replacement not only leads to an accessible contact distance so that gases can easily adsorbed and desorbed, but also prevents the target gas from being captured by stray particles, which may hinder gas detection. Pt functionalization by the galvanic reaction was performed after synthesizing the Cu_2_O nanopattern (Figure 1f). Since galvanic replacement generally results in a porous interior morphology [34], changing the exterior surface of the original material using another precursor in a strongly acidic environment induces fast reduction, and the reaction pathway can be controlled by precise adjustment of the solution pH [35]. To control the reaction time slowly and prevent collapse of the nanopattern structure by reducing the power of galvanic reaction moderate, a certain amount of sodium hydroxide (NaOH) solution was added. Under harsh acidic conditions, the synthesized Cu_2_O channels were destroyed, resulting in rapid dissolution of Cu_2_O. Generally, reduction by a galvanic reaction will lead to the formation of a hollow structure via oxidation of the inner metal [34]. When the Cu_2_O sensing array is dispersed in an aqueous mixture solution which are Pt^4+^ precursor for functionalizing, H^+^ to initiate the galvanic reaction and NaOH salt for limiting the chemical reaction slowly maintaining the pH value around 5, electrons transfer from Cu_2_O to Pt^4+^ and reduction of Pt^4+^ itself on Cu_2_O surface occur at the same time. Since Pt has a higher reduction potential energy than does Cu_2_O, Pt is preferentially reduced, so that Cu_2_O is replaced with Pt on the surface [36]. It is well known that when the pH of the reaction solution increases, the reducing power of AA is enhanced [35].

Therefore, by controlling the pH of the reaction solution with NaOH, we can adjust the galvanic reaction slowly, thus allowing both replacement and reduction without destroying the line pattern structure. After several hours of galvanic reaction at room temperature on a shaker, Pt ions were reduced into Pt particles onto the Cu_2_O array surface, resulting in a Pt-decorated Cu_2_O nanopattern (Figure 1g). To induce complete replacement, the experiment was performed in the same manner except for the addition of NaOH in the case of Pt nanowire array fabrication (Figure 1h). Under strongly acidic conditions without the addition of NaOH, the galvanic reaction predominates, resulting in complete replacement of Cu_2_O with Pt. To understand the reaction mechanism, schematic illustrations and supporting experimental results are presented in Appendix A.

### 3.2. Morphology, Elements, and Dimension Characterizations

To verify that selective Pt functionalization occurred only on the desired spot, scanning electron microscopy (SEM), energy-dispersive X-ray spectroscopy (EDS), and atomic force microscopy (AFM) were used (Figure 2). The pristine Cu_2_O nanochannel with high resolution and high aspect ratio was well fabricated over a large area with a periodic spacing of 500 nm (Figure 2a). X-ray diffraction (XRD) shows that Cu nanopattern is successfully synthesized to Cu_2_O nanopattern via thermal oxidation process (Appendix A). Photographs of the devices show that the Cu_2_O nanopattern was fabricated over a large area (cm^2^ scale) with excellent pattern fidelity (Appendix A). After 4 h of galvanic reaction, a part of the Cu_2_O channel was successfully replaced with Pt nanoparticles without destruction or collapse of the high-aspect-ratio nanostructure. The yellow particles on the Cu_2_O surface in Figure 2c were confirmed to be Pt via point EDS analysis by comparison that of the pristine Cu_2_O. In addition, the overall distribution of the functionalized Pt was observed by TEM mapping analysis (Appendix A). Low-magnification SEM-EDS images clearly confirmed that the Pt nanoparticles existed only in the Cu_2_O nanopatterns and there was no Pt in the substrate area between the Cu_2_O channels (Figure 2c–f and Appendix A). These results revealed that the galvanic replacement selectively occurs at the target spots, i.e., the surface of the Cu_2_O nanopatterns. With the reaction was carried out for a sufficient time under acidic conditions, replacement of Cu_2_O with Pt was predominant, resulting in a pure Pt line pattern at the site of the Cu_2_O pattern. To ensure complete reaction, the reaction was performed without the addition of NaOH (pH ~2). After a few hours, a Pt line pattern could be located at the original Cu_2_O channel spots (Figure 2e), which supported the occurrence of the galvanic reaction, while EDS observations confirmed the presence of pure Pt wires (Figure 2f). AFM analysis revealed the feature dimensions of the Cu_2_O nanopattern channel. The periodic Cu_2_O lines were 15 nm in width and 320 nm in depth before the galvanic reaction; after Pt doping, the Cu_2_O line nanopattern maintained its original features within the error range of the synthesis, thus indicating that the structure did not collapse during doping (Figure 1g–h). Overall, we fabricated high-resolution (15 nm) and high-aspect-ratio nanopatterns uniformly on a large area and achieved selective functionalization of Pt nanoparticles on the patterns. As discussed in a previous study [6], these nanopattern arrays have distinct advantages as a gas sensor: (1) polycrystalline feature with small grains; (2) low channel resistance caused by the large surface-to-volume ratio, which is derived from the ultrahigh resolution and high aspect ratio (~25). In addition, the noble metal acting as a sensitizer to enhance the gas sensing performance could be selectively decorated onto the surface of the ultrahigh-resolution nanopattern array by the galvanic replacement method.

### 3.3. Chemical Binding States of Pt/Cu_2_O Nanochannel

To confirm the chemical binding state and oxidation state of the nanopattern channel, detailed surface elemental analysis was carried out by X-ray photoelectron spectroscopy (XPS). The obtained binding energy was calibrated with that of the C-C peak, 284.7 eV. Overlapping peaks were deconvoluted and identified using the Avantage Software program (Thermo Scientific^TM^). As shown in Figure 3a, the metallic Cu nanopattern was completely oxidized and converted to Cu_2_O upon thermal annealing. For Cu 2p photoelectron analysis, the binding energy range from 925 to 965 eV was scanned. The binding energy of metallic Cu was not detected in the Cu 2p spectrum, confirming that the Cu channel was fully oxidized into Cu_2_O under the mild oxidation conditions. The high-resolution Cu 2p spectrum showed two main peaks at 933.18 eV and 952. 98 eV, which could be assigned to the Cu^2+^ double peaks for Cu 2p_3/2_ and Cu 2p_1/2_ and three satellite peaks, respectively [37]. The two O 1 s peaks represented the O^2-^ state (529.18 eV) from Cu_2_O and the OH^-^ state (531.17 eV) from the adsorption of atmospheric oxygen on the surface [38]. The Cu 2p and O 1s peaks confirmed the formation of metal oxide (Cu_2_O). Figure 3c shows the electron states of the Pt atoms in the nanopattern channel. The doublet peaks at 74.24 eV and 71.19 eV corresponded to the binding energy of metallic Pt, and the binding energies at 78.2 eV and 75.3 eV were in agreement with Pt^4+^ which came from the binding between metal oxide and doped Pt [39]. During galvanic replacement, the Pt^2+^ ions penetrated the Cu_2_O environment to form Cu_2_O/Pt species, ascribed to Pt^4+^ in the X-ray photoelectron spectrum. This Pt species binds to other Pt ions by accepting electrons from Cu_2_O to form metallic Pt. From the Pt 4f spectral analysis, we can confirm that Pt^2+^ in the galvanic reaction solution (PtCl_4_^2−^) is successfully reduced to Pt metal without remaining in the ionic state on the Cu_2_O surface.

The synthesized ultrathin Pt/Cu_2_O nanopattern exhibited 11-fold higher sensitivity than did the pristine Cu_2_O nanopattern. To determine the feasibility of using the Cu_2_O/Pt nanopattern as an electronic sensing channel, the baseline resistances and noise level of the channel were measured during 300 °C operation (Figure 4a). The channel resistance of the pristine Cu_2_O sensor was 24 MΩ, which increased, and approaching ~45 MΩ upon Pt nanoparticle decoration; a resistance level of tens of MΩ is within the optimal range for MOS-based gas sensors. The increase in the channel resistance was due to the formation of a depletion layer on the Cu_2_O channel because of electron transfer from the decorated Pt nanoparticles. In addition, the noise from the Cu_2_O/Pt channel increased from 0.88% to 4.59% of the baseline resistance. However, the Pt nanopattern obtained by the excessive galvanic reaction showed a significantly low resistance (tens of Ω) with an ultra-low noise level due to the metallic conductivity of the Pt nanopattern channel. To investigate the effects of Pt nanoparticle decoration on the sensing characteristics of the Cu_2_O nanopattern, a sensing device was fabricated with integration of a nanopattern channel (Cu_2_O, Cu_2_O/Pt, Pt) and two-terminal resistor type electrodes. A constant bias from 0.5 to 1.5 V was automatically applied to the two-probe electrode, and the electrical resistance of each channel was recorded as a sensing signal by a data acquisition module (Agilent 34970A). The sensing devices were simultaneously loaded on a home-made gas sensing chamber, and the sensing signals from each device were measured with multi-channel sensing systems. 

### 3.4. NO_2_ Sensing Performances and Mechanism

Details of the in-house fabricates gas delivery system are provided in the Supporting Information (Appendix A) [40]. Figure 4b shows the responses of the Cu_2_O, Cu_2_O/Pt, and Pt nanopatterns toward 100 ppm NO_2_ at 300 °C. Gas response is defined as R_a_/R_g_, where R_a_ and R_g_ are the resistances of the sensor in air and in the target gas, respectively. The Cu_2_O/Pt nanopattern exhibited a markedly higher sensitivity upon exposure to NO_2_ (R_a_/R_g_ = 11) than did the pristine Cu_2_O nanopattern (R_a_/R_g_ = 1), which was attributed to the effect of the noble metal (Pt) nanoparticle decoration. For the Pt nanopattern obtained by the excessive galvanic replacement reaction, NO_2_ detection was not observed due to the perfect metallic properties of the nanopattern. To further investigate the high sensitivity of the Cu_2_O/Pt nanopattern sensor, real-time sensing response to single ppm of NO_2_ (1 to 4 ppm) is measured (Figure 4c). It is clearly seen that Cu_2_O/Pt nanopattern sensors show significant responses to single ppm NO_2_ with distinguishable signal resolutions. Figure 4d further demonstrates that the Cu_2_O/Pt nanopattern sensors show linear response variations to single ppm NO_2_ (1–4 ppm) with significant repose amplitudes ((ΔR/R_b_)_max_) which was comparable value with other previous works (Appendix A). Here, R_b_ and ΔR represent the baseline resistance of the sensor exposed to dry air and the change in resistance after exposure to NO_2_, respectively. Thus, selective functionalization of the Cu_2_O channel by galvanic replacement had a significant impact on the enhancement of the gas sensing performance. The response/recovery time (τ90%, time taken to reach the 90% of the minimum resistance level) of the sensors for NO_2_ is shown in Appendix A. Moreover, functionalized Cu_2_O/Pt doesn’t change the selectivity characterization comparing that of Cu_2_O (see the Appendix A which shows similar response selectivity comparing that of Cu_2_O [6] among various gases). In addition, long-term stability test of the Cu_2_O/Pt nanopattern sensor was conducted showing high-stability of synthesized catalyst (Appendix A). The highly enhanced gas sensing performance of Cu_2_O with the Pt galvanic reaction could be explained via two mechanisms: electronic sensitization (ES) [24,41,42] and chemical sensitization (CS) [24,43,44]. The detailed enhancement mechanisms are shown in Figure 4e–g. First, the decorated Pt nanoparticles played an ES role during the sensing process. When exposed to air, each Pt nanoparticle consisting of Pt and Pt^4+^ forms a redox electrode Pt^4+^/Pt^0^. To achieve electronic equilibrium with the Pt nanoparticles, the Fermi level of Cu_2_O will be shifted and pinned at the Pt^4+^/Pt^0^ electrode potential, which produces an electron-depleted space charge region and decreases the charge (hole) accumulation layer (HAL) (Figure 4e). This eventually leads to an increase in the resistance of the pristine Cu_2_O/Pt in air. Second, it the functionalized Pt nanoparticles also play an important role via the CS mechanism because of the spillover effect. For the pristine Cu_2_O nanopattern, the adsorbed oxygen ions (O^−^, O^2−^) localize the electrons, which leads to an increase in the concentration of holes in the surface layers. Upon exposure to NO_2_, the adsorbed NO_2_ ions (NO^2−^) further localize the electrons of Cu_2_O, thus increasing the HAL thickness and decreasing the resistance of the sensors (Figure 4f–g). In the case of Cu_2_O/Pt, significantly larger amounts of NO_2_ can be adsorbed onto Cu_2_O via the spill-over effect [43,44] due to the high catalytic activity of the Pt nanoparticles (Figure 4f). Thus, the high-resolution Cu_2_O nanopattern is selectively functionalized with Pt nanoparticles via the galvanic replacement reaction, and the gas sensing performance is significantly improved via both the ES and CS mechanisms.

## 4. Conclusions

In conclusion, an ultrahigh resolution array, i.e., a thin, polycrystalline, and high-aspect-ratio p-type sensing channel array (Cu_2_O) was selectively functionalized with Pt via galvanic replacement. Herein, by using a galvanic reaction which involves a chemical reaction between two materials, we achieved selective doping of Pt onto the surface of the ultrahigh-resolution nanopattern array. We successfully fabricated an ultrathin nanopattern array with a high surface-to-volume ratio over a large area by our unique lithography technique (low-energy plasma bombardment [23]) and showed enhancement of the sensing performance after selective Pt functionalization through the galvanic replacement reaction. The sensor with Pt-functionalized Cu_2_O showed a 11-fold increase in the gas response (R_a_/R_g_) to the bare Cu_2_O channel when exposed on 100 ppm NO_2_. The role of the functionalized Pt particles was suggested in terms of both ES and CS mechanisms. Using the selectively Pt-decorated high-resolution Cu_2_O array obtained via galvanic replacement, we could achieve significant enhancement of the NO_2_ sensing performance.

## Figures and Tables

**Figure 1 sensors-18-04438-f001:**
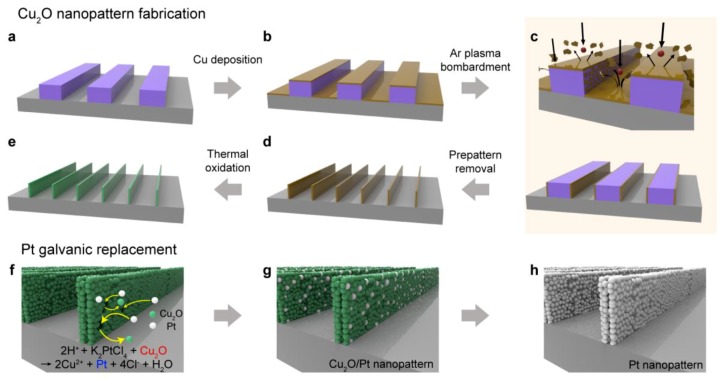
Schematic illustration of fabrication of Pt-functionalized high-resolution Cu_2_O nanopattern. (**a**) The PS prepattern was transferred from the PDMS mold. (**b**) The target metal (Cu) was deposited onto the prepattern. (**c**) By low-energy plasma bombardments, the deposited metal was widely sputtered onto the side wall of the prepattern, resulting in polycrystalline Cu walls. (**d**) The prepattern residue was removed by RIE and (**e**) the ultra-thin Cu line pattern was oxidized by thermal annealing. (**f**) To functionalize the Cu_2_O surface with Pt, the prepared channel was immersed in a Pt precursor solution. (**g**) The electrons were transferred from Cu_2_O to Pt ions, resulting in the Pt-decorated Cu_2_O nanopattern. (**h**) In case of an excessive galvanic reaction, the Cu_2_O baseline channel was completely converted into Pt.

**Figure 2 sensors-18-04438-f002:**
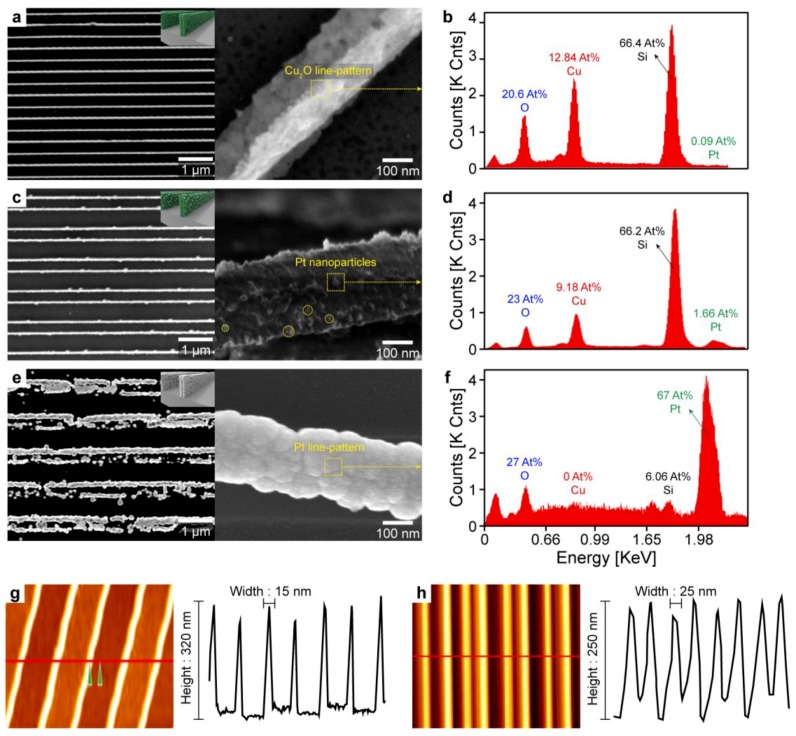
Characterization of fabricated Cu_2_O, Pt/Cu_2_O, and Pt sensing channel. SEM image of the fabricated pattern array for (**a**) pristine Cu_2_O, (**c**) optimized Pt-doped Cu_2_O, and (**e**) completely reacted Pt. (**b**–**f**) EDS point analysis of each channel. (**g**–**h**) AFM depth and width profiles show ultrathin line channel and no disruption of the pattern after Pt doping.

**Figure 3 sensors-18-04438-f003:**
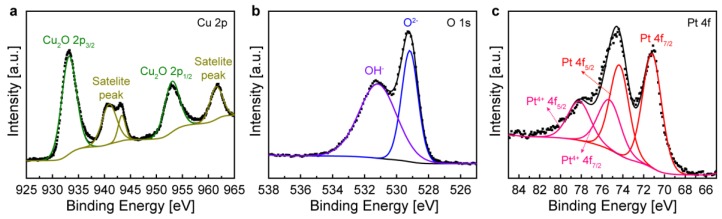
X-ray photoelectron spectra of Cu_2_O/Pt nanopattern. High-resolution elemental spectra of (**a**) Cu 2p, (**b**) O 1s, and (**c**) Pt 4f.

**Figure 4 sensors-18-04438-f004:**
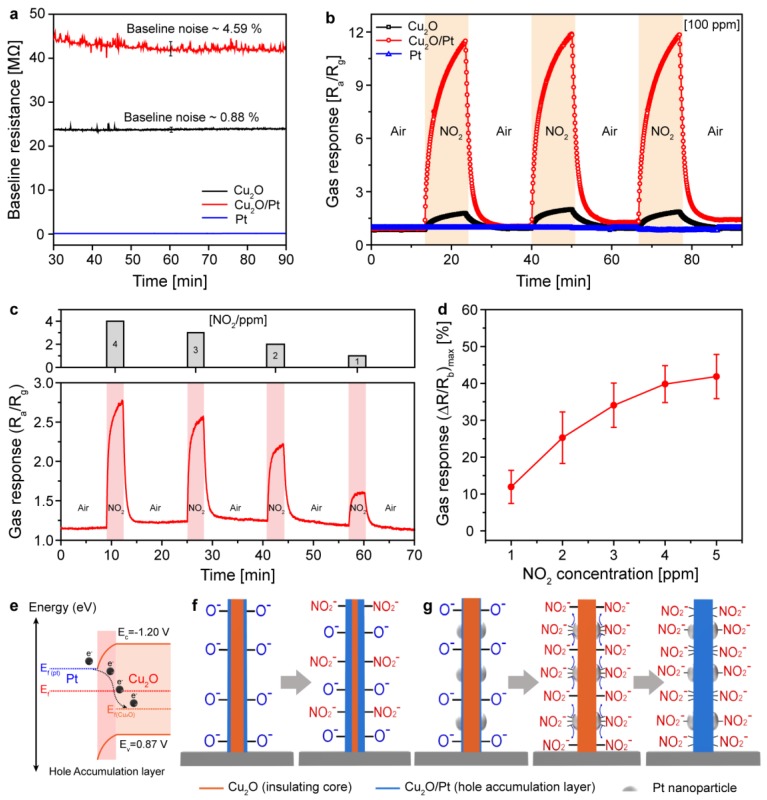
Device characteristics and gas sensing performance of the Cu_2_O, Cu_2_O/Pt, and Pt nanopattern sensors. (**a**) Baseline resistance and noise of the Cu_2_O, Cu_2_O/Pt, and Pt nanopattern. (**b**) Real-time NO_2_ (100 ppm) sensing performance of the Cu_2_O, Cu_2_O/Pt, and Pt nanopattern sensors. (**c**) Real-time response behavior of the Cu_2_O/Pt nanopattern sensor to low concentration NO_2_ (1 to 4 ppm). (**d**) Maximum response amplitudes of the Cu_2_O/Pt nanopattern sensor (three sensors) to low concentration NO_2_. (**e**) Band diagram scheme of Cu_2_O/Pt to elucidate electronic sensitization (ES). Schematic illustrations of chemical sensing mechanism of (**f**) Cu_2_O and (**g**) Cu_2_O/Pt nanopattern with chemical sensitization (CS) of spill-over effect by Pt nanoparticles.

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
