# Peer review of "Selective Functionalization of High-Resolution Cu_2_O Nanopatterns via Galvanic Replacement for Highly Enhanced Gas Sensing Performance"

_sensors, 2018, doi:10.3390/s18124438_

Reviewer 1 Report

General note: The Abstract should not contain the Introduction and references, but should briefly reflect which studies were carried out in this work and which main results were obtained.

The article is of particular interest from a technological point of view. However, for readers working in the field of materials for gas sensors, it is necessary to clarify the following points.

1. It is necessary to detail the description of the sensor measurements procedure. What are the gas flows in the measurements? How was the concentration of NO2 100 ppm created? What was the source of NO2 (certified gas mixture or something else)? What was the background gas (air or nitrogen)? Fig. S6 is misleading regarding the concentration of NO2 and background gas.

2. How was the sensor heated to 300C? Why was such a high temperature chosen, if it is well known that for detecting NO2 with semiconductor oxides, the optimum temperature is 150–200C, and with a further increase in the measurement temperature, the sensor signal decreases significantly?

3. XPS - The authors attribute the signal in the Cu 2p spectrum to Cu (I), but then postulate the formation of copper oxide (I) Cu2O. Why? In fact, the signals of Cu (1) and Cu (2) in the XPS spectra are very close and unambiguous interpretation is difficult. What is evidence that exactly Cu2O is formed? It is necessary to make the additional studies by X-ray diffraction.

4. What is ΔR/Rb? It is not defined.

5. To clarify the text in lines 283-286, it is necessary to provide a diagram of the band structure of the Cu2O/Pt heterocontact.

6. The sentence "the adsorbed oxygen ions (O- , O2-) attract the hole carriers of Cu2O" in line 288 is wrong. During chemisorption on the surface of semiconductor oxides, oxygen localizes electrons, which leads to an increase in the concentration of holes in the surface layer.

7. Similarly, in line 290 for the adsorption of NO2.

8. Line 292 - confirmation of NO2 spillover is required (from literature data) .

Author Response

I attached the response file (Please see it)

Reviewer 2 Report

General Comment: The manuscript entitled “Selective Functionalization of High-Resolution Cu2O Nanopattern via Galvanic Replacement for Superior Gas Sensing Performance”. The manuscript is well written and the results are interesting, however, it requires some refinement before the publication. The manuscript can and should be further improved by taking the following aspects into consideration.

1.     The author did air anneal at 450C? What is the reason to choose the specific temperature? It would be good to show the TGA plot before annealing if the crystallization is the main reason for the annealing.

2.     Include XRD plot before and after annealing of the sample.

3.     In the experiment part, please include fabricated sensor image.

4.     Page 2, (Introduction section). Author has mentioned that “Recently, a patterned p-type polycrystalline” Some recent reports should be cited instead of outdated or earlier papers. Take a look and include these references in introduction part and for P-type gas sensing mechanism: ACS Appl. Mater. Interfaces 2014, 6, 16, 13917-13927, RSC Adv., 2016, 6, 92655, J. Mater. Chem. A, 2018,6, 17120-17131, Sensors and Actuators B 257 (2018) 906–915, Sensors and Actuators B 153 (2011) 347–353

5.     Fig. 4(b):  Authors have not measured steady-state response properties.  Please re-measure all data.

6.     Please provide response and recovery time Vs concentration with error bar.

7.     Please provide comparison table with other metal oxide and metal oxide heterostructures which shows that this sensor is better from the technology point of view.

8.     Include selectivity histogram of pure and Pt-modified Cu2O samples toward different gases.

9.     It is better to include some strategies to improve selectivity and cite relevant references: Some suggested review article to cite in the intro. Part: Adv. Mater. 2016, 28, 795–831, Microchimica Acta, 185(2018) 213, Adv. Funct. Mater. 2017, 27, 1702168.

10.  What is the stability of the sensor, please include long-term stability of sensor performance for at least 30-60 days.

11.  The quality of some figures needs to be enhanced.

12.  It is better to check and correct the font size of x and y-axis of all figures in the manuscript. It should be same.

Author Response

(The authors gave the same response as above.)

Reviewer 3 Report

In this paper authors presents the nanopaterned Cu2O structure funtionalized by Pt (using galvanic reaction) for gas (NO2) sensing aplication. 

From the material science point of view this paper is very interesting and novel. It is continuation of authors previous paper published in Nano Letters journal (ref [1]). Authors presetned very interesting rute of obtain Cu2O/Pt structure and characterized it using FE-SEM, TEM, EDS, AFM and XPS methods. This is strong side of the paper.

However, from the gas sensing point of view paper have to be extensively improved. First of all, authors presented response only to 100 ppm of NO2 at 300oC. It is not an outstanding result for MOX sensors. Such concentration is a high concentration for this gas because from the practical point of view it should be monitored in the level o single ppms or even ppb/ppt level. Also the temperature of operation of this sensor is not very atractive because nowadays  rather low temperature (most of all room temperature) operating sensors are investigated. So authors should carry on an experiment with lower NO2 concentrations (there is potential for that). They also should show how sensor response is changing with the concentration (please show data with diffrend concentrations) to show if sensor is scalable.

From the figures presented by authors (Figs 4 b,c and d) it can be noticed that for structure functionalized by Pt response is significanly higher than in the case of clean Cu2O one, but response time seems to be very similar or equal (fig 4d) and the recovery time is significanlty higher (fig 4c) for Pt/Cu2O structure. Authors had written in few places of the manuscript (abstract, rults and discussion, conclusions) that the funtionalized sensor is faster, but the results are showing something else...

What is more authors do not prepare the gas sensing experiments properly... Please take a look in the Fig 4b - the sensing signals do not achieved the stady state after the 10 min of the NO2 exposition (It can be seen also on fig 4d) in the case of the Pt/Cu2O structure also during the sensor recovery signal do not falls to the stable value (Fig 4c). This is why the experiment with gas have to be conducted once again with longer cycles times (30 min or 1h?).   

Fig 4d do not have caption.

Why atuthors do not show the raction to other gases. The presence of Pt in the structure at such temperature (300oC) should have good response to for example hydrogen. Authors also should check the selectivity of the sensor for some other chosen gases. 

From above it can be seen that adjective "Superior" used in the title is not a good idea. Please compare your results to other papers to show if the obtained structure is better or comaprable with other materials proposed in literature for NO2 sensing.

The sensing mechanism is described in the manusctipt without any references to literaure or experimentl proofs - please try to prove that this description is proper. 

In my opinion in abstract should not be used references and information form previous works. For example information about response to hexane  in the abstract is misleading. As the abstract appears often separately from the  full paper using of references in there is rather not a good idea.  

In the introduction only 4 or 5 literture references are published after 2015, to show current state of the art properly please use more actual literature if possible. 

In summary, at this moment paper shows the rute to obtain an intersting sensing structure and proves that funtionalization improves its reaction to quite high NO2 concentration in quite high temperature. So, it is interesing from material science point of view, but from gas sensing point of view authors have to prove its value to publish the paper in this journal. This is why, in my opinion, paper can be reconsider after major revision.

Author Response

I attached the response file (Please see it)

Round  2

Reviewer 1 Report

The article can be accepted in its present form and can be published after standard procedure of English editing and proofreading.

Reviewer 3 Report

The authors addressed all my comments and revised the manuscript in satisfactorily way. From my point of view paper can be published in the current form.